# Synthesis and Cytostatic Effect of 3’-deoxy-3’-*C*-Sulfanylmethyl Nucleoside Derivatives with d-*xylo* Configuration

**DOI:** 10.3390/molecules24112173

**Published:** 2019-06-10

**Authors:** Miklós Bege, Alexandra Kiss, Máté Kicsák, Ilona Bereczki, Viktória Baksa, Gábor Király, Gábor Szemán-Nagy, M. Zsuzsa Szigeti, Pál Herczegh, Anikó Borbás

**Affiliations:** 1Department of Pharmaceutical Chemistry, University of Debrecen, 4032 Debrecen, Egyetem Tér 1, Hungary; bege.miklos@pharm.unideb.hu (M.B.); kicsak.mate@pharm.unideb.hu (M.K.); bereczki.ilona@pharm.unideb.hu (I.B.); herczegh.pal@pharm.unideb.hu (P.H.); 2Department of Biotechnology and Microbiology, University of Debrecen, 4032 Debrecen, Egyetem Tér 1, Hungary; kissalexandra0329@gmail.com (A.K.); viktoriabaksa@gmail.com (V.B.); kiru.eger@gmail.com (G.K.); mzsuzsa768@gmail.com (G.S.-N.); bigdegu@gmail.com (M.Z.S.)

**Keywords:** xylofuranosyl nucleoside, cytostatic, genotoxic, squamous carcinoma cell line (SCC), radical thiol-ene coupling, time-lapse imaging video-microscopy

## Abstract

A small library of 3’-deoxy-C3’-substituted xylofuranosyl-pyrimidine nucleoside analogues were prepared by photoinduced thiol-ene addition of various thiols, including normal and branched alkyl-, 2-hydroxyethyl, benzyl-, and sugar thiols, to 3’-exomethylene derivatives of 2’,5’-di-*O*-*tert*-butyldimethylsilyl-protected ribothymidine and uridine. The bioactivity of these derivatives was studied on tumorous SCC (mouse squamous carcinoma cell) and immortalized control HaCaT (human keratinocyte) cell lines. Several alkyl-substituted analogues elicited promising cytostatic activity in low micromolar concentrations with a slight selectivity toward tumor cells. Near-infrared live-cell imaging revealed SCC tumor cell-specific mitotic blockade via genotoxicity of analogue **10**, bearing an *n*-butyl side chain. This analogue essentially affects the chromatin structure of SCC tumor cells, inducing a condensed nuclear material and micronuclei as also supported by fluorescent microscopy. The results highlight that thiol-ene chemistry represents an efficient strategy to discover novel nucleoside analogues with non-natural sugar structures as anticancer agents.

## 1. Introduction

Nucleoside analogues play pivotal roles in antiviral [1,2,3,4] and anticancer [4,5] chemotherapy. They are chemically modified analogues of natural nucleosides, which are endogenous compounds involved in many essential cellular processes, such as DNA and RNA synthesis, cell signaling and metabolism. Currently, there are more than 10 approved nucleoside derivatives used to treat various cancers, and many other nucleoside analogues are being investigated in clinical trials. Most of these derivatives act as antimetabolites, compete with natural nucleosides, and interact with a large number of intracellular targets inducing cytotoxicity. 

A number of the cytotoxic and antiproliferative nucleoside analogues featured modifications to the sugar unit including addition or removal of substituents on the furanose ring, inversion of the C2’ and C4’ configurations or changing the furanose oxygen into carbon, sulfur or nitrogen (Figure 1). Some examples for 2’-modified nucleosides approved for the treatment of hematological malignancies include cytarabine (1-β-d-arabinofuranosylcytosine), fludarabine (9-β-d-arabinofuranosyl-2-fluoroadenine), clofarabine [2-chloro-9-(2-deoxy-2-fluoro-β-d-arabinofuranosyl)-adenine], and nelarabine (2-amino-9-β-d-arabinosyl-6-methoxy-guanine), which are arabinose analogues with an inverted configuration at the C2’ position [6,7] (Figure 1A). A further example for the most common 2’ modification, gemcitabine (2′-deoxy-2′,2′-difluorocytidine) containing a C2’-geminally difluorinated furanosyl unit is used for the treatment of various types of cancers [8,9,10]. Capecitabine (5′-deoxy5-fluoro-*N*-[(pentyloxy)carbonyl]cytidine), possessing a 5’-deoxy sugar unit is a third-generation prodrug of 5-fluorouracil approved for treatment of metastatic colorectal and breast cancers [11]. Forodesine is an imino-*C*-nucleoside purine derivative, in which the furanose ring oxygen is changed to nitrogen, approved in Japan in 2017 for the treatment of refractory peripheral T-cell lymphoma [7]. 

Although currently there is no 3’-modified nucleoside among the approved anticancer drugs, several 3’-deoxy or 3’-branched derivatives possess remarkable anticancer activity (Figure 1B). The active structures lacking the 3’-OH group include cordycepin, 3’-deoxy adenosine, which is a natural nucleoside derivative of fungus origin [12] and troxacitabine, β-l-dioxolane-cytidine, the first l-nucleoside which was studied as an anticancer agent [13]. Troxacitabine showed promising activity against solid tumors and leukemias [14,15,16] and cordycepin proved to be effective for the treatment of refractory TdT-positive leukemia [5,17]. Some 3’-*C*-alkylated analogues such as 3’-*C*-methyladenosine [18,19,20] and 3’-*C*-ethynylcyidine (TAS-106) [21,22] served as potent anticancer agents against numerous human leukemia and carcinoma cell lines. Further study with regioisomers of 3’-*C*-methyladenosine revealed that shifting the methyl from the 3’ carbon to another position on the sugar ring led to decrease in activity, thus highlighting the importance of the 3’-modification [19]. Very recently, 3’-*C*-ethynyl-β-D-ribofuranose 7-deazapurine nucleosides were found to elicit potent antiproliferative activity against several solid tumor derived cell lines [23,24]. Importantly, these 3’-branched derivatives, being ribonucleosides, can target both DNA and RNA, therefore they were hypothesized to offer a broader interval in the cell cycle to exert anticancer effects [25].

Despite the availability of several anticancer drugs and the large number of the identified anticancer nucleoside analogues, there is a constant need for development of newer analogues with enhanced chemical and biological properties to overcome issues of resistance and long-term toxicity. 

Recently, we have investigated the photoinduced thiol-ene reaction, also known as thio-click reaction [26,27,28,29,30], as a novel strategy for the synthesis of sugar-modified nucleosides [31]. We have found that the addition of 1-propanethiol and 1-thiosugars to 3’-exomethylene derivatives of uridine and ribothymidine at −80 °C proceeded with high efficacy and almost exclusive d-*xylo* stereoselectivity. The obtained d-*xylo* configured nucleoside derivatives (Figure 2), containing a thio-substituted 3’-*C*-methyl moiety in the “up” position of the sugar ring, show very close structural resemblance to the 3’-*C*-methyl- [18,19,20] and 3’-*C*-ethynyl-nucleosides [21,22,23,24] which exhibit potent anticancer activity. Thus, we hypothesized that our new analogues might possess cytotoxic or antiproliferative activity. Because of the mild conditions, atom economy and broad range of commercially available thiols, the thiol-ene coupling reaction offers a convenient and straightforward route to couple various substituents to a core molecule. Therefore, we decided to exploit the thio-click chemistry to produce a series of 3′-deoxy-3′-*C*-substituted nucleosides for studying their anticancer activity. 

## 2. Results and Discussion

### 2.1. Chemistry

First, as a starting point of this research, the cytotoxic effect of the 3′-modified uridine (**1** and **2**) and ribothymidine (**3–5**) derivatives available from our recent work [31] was determined on squamous cell carcinoma (SCC) and control HaCaT (human keratinocyte) cell lines (Figure 2). While the 1-thiosugar-modified derivatives **1, 3** and **4** showed no cytotoxicity (except for **3** which was quite toxic to the control healthy cells), the 3’-C-propylsulfanylmethyl uridine and ribothymidine analogues **2** and **5** exerted remarkable toxic effect on both the tumorous and control cell lines. Fortunately, in the case of **5** a slight selectivity toward the tumor cells was observed. On the basis of these data, compound **5** was chosen as the lead structure to produce novel 3′-C-modified nucleoside analogues.

The starting exomethylene derivatives **7T** and **7U** were synthesized by oxidation-methylenation protocol of 2’,5’-di-O-*tert*-butyldimethylsilyl-ribothymidine **6T** and 2’,5’-di-O-tert- butyldimethylsilyl-uracil **6U**, as reported previously [31] (Scheme 1). In order to study the effects of the lipophilicity, aromaticity, and bulkiness of the 3’-substituent on the cytotoxicity, the ribothimidine derivative **7T** was reacted with a panel of commercially available thiols. The applied thiol reagents include normal alkyl thiols with a length of C2-C12 (ethyl-, *n*-propyl-, *n*-butyl-, *n*-hexyl-, *n*-octyl-, and *n*-dodecyl mercaptane), branched alkyl thiols (*i*-propyl, *i*-butyl and *t*-butyl-mercaptane), phenyl- and benzyl-thiols, as well as 2-hydroxyethyl mercaptane (Table 1, entries 1–15).

Recently, we have demonstrated that the low reaction temperature has a significant beneficial effect on both the efficacy and stereoselectivity of the hydrothiolation of nucleoside exomethylene derivatives [31]. Accordingly, the thiol-ene reactions of **7T** were initially carried out at −80 °C under the previously established conditions, using UV-irradiation (λ_max_ = 360 nm) in the presence of the photoinitiator 2,2-dimethoxy-2-phenylacetophenone (abbreviated as *DMPA or DPAP*) [31,32]. While the reaction with ethyl- and 2-hydroxyethyl mercaptane proceeded with high yields (Table 1, entries 1 and 13), the higher alkyl homologues and the aromatic thiols showed lower or no reactivity at −80 °C. We assumed that the longer and branched alkyl chains stabilize the electrophilic thiyl radical in a higher extent, in line with their electron-donating capability, thereby shifting the equilibrium of the rapidly reversible addition of the thiyl radical onto alkenes toward starting compounds [32,33]. Therefore, in order to increase the reactivity of the intermediate thiyl radicals, the reactions were carried out at −40 to 0 °C, depending on the reaction progress monitored by TLC (thin layer chromatography). This way, fair to good yields were observed in all cases except for thiophenol, which failed to react with alkene **7T** (Entry 11). Such failure with thiophenol was already reported and explained by the high resonance stabilization of the aromatic thiyl radical [33]. In general, the reactions proceeded with high to excellent d-xylo selectivity, albeit the lower sterical demand of thiols (entries 1, 2 and 12) and the higher temperature (entries 5 vs. 4 and 18 vs. 17) resulted in a lower level of stereoselectivity. Interestingly, the stereoselectivity observed with 2-mercaptoethanol was only modest and showed an opposite temperature dependence than that of the apolar thiols (entries 13–15). In order to provide uracil-containing reference compounds for the cytotoxicity study, the uridine derivatives **7U** was reacted with *n*-propyl and *n*-butyl mercaptanes (Table 1, Entries 16–18). The reactivity and stereoselectivity trends of these additions were the same as observed with the thymidine congener **7T**, apart from that the lack of the methyl substituent at position C5 resulted in a significantly higher level of diastereoselectivity.

The structure and diastereomeric ratio of the new compounds were determined on the basis of their ^1^H and ^13^C NMR spectra, using the d-xylo configured derivative **3** and its C3’-epimer **3**-**d-ribo** as the reference compounds [31]. The characteristic ^1^H and ^13^C NMR chemical shifts and ^3^J_H,H_-coupling constants of the major d-xylo isomer of each product are listed in Table 2. The characteristic spectral data of the major components of **8–18**, (H1’-H2’ coupling constant, H-4’chemical shift as well as the C1’ and C4’ chemical shifts) showed very high similarity to the corresponding data of the d-xylo-configured reference compound **3**.

Continuing the synthesis towards the deprotected nucleoside analogues, the ethylthiomethyl-substituted derivative **8** was desilylated with tetrabutyl ammonium fluoride to give **19** in 30% yield (Scheme 2). As compound **19** did not show any cytotoxic activity (see Table 3), further deprotected derivatives were not prepared.

### 2.2. Biological Evaluation

#### 2.2.1. Cell Viability Study

For the investigation of the cytotoxic effect of nucleoside analogues we used healthy human keratinocyte (HaCaT) and tumorous squamous carcinoma SCC-VII cell lines. In the experiments, a standard citotoxicity assay, the MTT assay, was used in a 96-well plate. Importantly, different cell numbers were used in the case of the two different cell lines. The HaCaT cells were seeded 10^4^/well concentration while the SCC-VII cells were seeded 5000/well concentration on the plate. In both cases the cells were treated with the nucleoside analogues at ~50% confluence [34]. We applied the different starting conditions to compensate for the faster generation time and consequently increased protein mass of the tumorous cells [35].

The half maximal inhibitory concentration (IC_50_) values of compounds **1–5** and **8**–**19** are summarized in Table 3. The protected derivatives, except for the sugar- and *n*-dodecyl-substituted compounds, inhibited cell growth of both cell lines in a concentration-dependent manner (see Appendix A in Appendix A) with an IC50 values of 10–30 µM. Within the group of active analogues, compounds **5**, **9**, **10**, **13** and **18** bearing an apolar and not too bulky substituent at 3′ position (substituents of C3-C6 length) displayed the most promising activity profile (Figure 2). They showed a very slight but consequent selectivity to the tumor cells with IC_50_ values of 11–17 µM and selectivity indexes (SI) of 1.32–1.64 (these compounds are located to the right of the diagonal in Figure 2). Interestingly, the uridine analogues **2** and **18** possess higher cytotoxicity but lower selectivity compared to their thymidine congeners **5** and **10**. 

Increasing the bulkiness/chain-length of the side chain to *t*-butyl (**12**) and *n*-octyl (**14**) leads to a drop in the cytotoxicity, and coupling as bulky substituents to the 3’-position as *n*-dodecyl (**15**) or acetylated sugars (**1, 3** and **4**) results in complete loss of activity. 

Surprisingly, the unprotected compound **19** showed no activity against either the healthy or the cancer cells (Table 3, entry 17). One possible reason for the complete inactivity might be the inefficient cellular uptake of **19**, possibly caused by the lack of 3’-OH group. It has been shown that the hydroxyl group at position 3’ is required for efficient uptake of nucleoside analogues by nucleoside transporters, as the 3’-deoxy compounds, such as cordycepin, are not good substrates of the transporter enzymes [5,37,38]. At the same time, introduction of a hydrophobic substituent to the 5’-position is a proven strategy to facilitate the cell penetration by passive diffusion [39,40]. For example, elacytarabine (CP-4055), a 5’-oleic acid ester derivative of cytarabine, was designed to enter cancer cells independently of nucleoside transporters [40]. In our case, the 5’-silyl substituent of **1–5** and **8–18** might play a role in entering the molecules to the cell by a transporter-independent manner.

Apart from the beneficial effect on the cellular uptake, the silyl groups might play an important role in the biological activity. There are numerous examples in the literature regarding the positive effect of silyl substituents on cytotoxic or antiviral activity of nucleoside analogues. For example, Herczegh and colleagues have demonstrated that the TBDMS-protected leinamycin-nucleosides showed higher growth inhibitory activity toward human cancer cells than the corresponding deprotected derivatives [41]. Bis-ureidoadenosine derivatives, reported by Peterson’s group, were found to display a higher antiproliferative activity in the protected form, and the silylated compounds showed the highest activity [42,43]. Moreover, in the case of the ribothymidine-derived reverse transcriptase inhibitor TSAO-T, the *tert*-butyldimethylsilyl group at 5’ positions is found to play a crucial role in the biological activity [44]. Very recently, Otterlo’s group reported that introduction of silyl groups to nucleosides afford compounds with interesting cytotoxic [45] and antiproliferative [46] activity. 

The well-demonstrated activity of simple silylated nucleosides [45,46] prompted us to study the cytotoxicity of the 2’,5’-disilyl derivative **6T**, which we used as a starting compound of this study (Table 3, entry 18). This unmodified nucleoside displayed remarkable toxicity against both cell lines, however, showing higher toxicity toward the healthy HaCaT cells. These results, in one hand, confirm the important role of the silyl substituents in the biological activity, and, in the other hand, highlight that an apolar side chain at 3’ position might confer selectivity towards the tumor cells. 

In order to better understand the mode of action of our nucleoside analogues, compound **10** having the most promising activity profile was selected for further studies.

#### 2.2.2. Live Cell Imaging via Time-Lapse Microscopy

Dynamic functional and morphological effects of compound **10** was investigated via time-lapse (1 frame/min) videomicroscopy under standard cell-breeding conditions. Low intensity, near-infrared (NIR) 940 nm illumination was used, for reduced phototoxicity. Parallel experiments were done in the same CO_2_ incubator, using two custom-built inverted microscope equipped with sensitive charge-coupled device (CCD) sensors. Using this Time-Lapse System (TLS), we studied the effect of compound **10** in 17.0 µM concentration, which is the half maximal inhibitory concentration (MIC_50_) of **10** against SCC cells. Image sequences were processed and quantitatively analyzed using NIH ImageJ open-source software-bundle. Cell size, cellular generation time, and cell growth of HaCaT and SCC cells were analyzed. Time-lapse video-microscopy record about the effect of 10 and DMSO control on HaCaT and SCC cell lines are available in Appendix A.

##### Mother Cell Size Changes 

The increased size of the mother cells, detached and rounded just prior to division, is a common indicator of cytotoxicity. 

Each cell line was treated with 17.0 µM solution of compound **10,** dissolved in 1% (*v*/*v*) DMSO and diluted with cell culture media before application. Then, 1% (*v*/*v*) DMSO was used as control. 

Comparison of HaCaT and SCC mother cells sizes reveals a moderate cytotoxic effect of DMSO and **10** (Figure 4). The treated SCC cells showed a significant (p < 0.05, ANOVA, multiple alignment) 12.8% increase compared to the DMSO (Figure 4). This relationship had not been observed on HaCaT cells. On HaCaT, the DMSO was more toxic than **10** (p < 0.05) and in fact, a 9% size-reduction was measured upon **10** treatment vs. DMSO (p < 0.05) (Figure 3). A ~21% overall size change difference was observed due to the treatment with compound **10** between the two cell lines. (Table 4, Figure 4).

##### Generation Time

For the investigation of the effect of compound **10** on cell-cycle, cellular generation times were measured from time-lapse image sequences. 

The HaCaT cell line generation time was slightly affected by DMSO treatment alone, but generation time elongated by 20.8% upon **10** treatment. 

Fast-dividing tumor cells, like SCC are characterized by rapid cell-cycle. Generation time of the SCC line is 67.4 % faster than non-tumorous HaCaT. DMSO treatment of SCC elongated the cell-cycle only slightly, similar to HaCaT. In contrast, upon treatment with compound **10**, the SCC generation time was not measurable within 48 h, since de novo daughter cells were unable to divide within the time-window of live-cell imaging (Table 5, Figure 5).

##### Growth Inhibition

Cellular monolayer growth in the log phase is the most evident indicator of cellular proliferation. Testing with HaCaT cell culture, the monolayer expansion rates were similar in the case of DMSO and compound **10** treatment (Figure 6). 

The SCC growth was significantly slower after treatment with compound **10** compared to DMSO, despite the 67.4% faster cell-cycle of SCC than HaCaT. Furthermore, the **10**-treated SCC cultures showed signs of division senescence after the first division of mother cells, suggesting a cell-cycle arrest after the mitosis-related cellular incorporation of compound **10**. (Note the lower starting number of SCC cells on Figure 6. Otherwise, untreated fast dividing SCC monolayer would reach 100% confluency before 48 h, exhibiting a contact-induced senescence.)

#### 2.2.3. Fluorescent Microscopy

In our experiments 4’,6-diamidino-2-phenylindole (DAPI) fluorescent DNA dye has been used for studying the cell nucleus morphology as an indicator of genotoxicity. The HaCaT and SCC cells were treated with DMSO and compound **10**, fixed and stained with DAPI, then examined with fluorescent microscopy (Figure 7 and Figure 8).

For DMSO controls, the effect of 1% (*v*/*v*) DMSO on nuclear morphology is negligible for both cell types. 

There is no apparent difference between the structure of the control and the treated nucleus in HaCaT, altough necrotizing cells were observed sporadically in DMSO samples (Figure 7).

The morphology of SCC cells nuclei changed upon treatment of compound **10**. Strong heterochromatinization is characteristic. Lack of mitotic nuclei suggests a tendency of senescence. Micronucleus formation is apparent, and nuclear swelling is also present. All these changes indicate the genotoxic effect of compound **10** on SCC (Figure 8).

## 3. Conclusions

A new type of cytostatic pyrimidine nucleoside analogues was identified, possessing a 3’-deoxy-3’-C-sulfanylmethyl-d-xylofuranosyl moiety. These compounds, obtained by photoinduced thiol-ene reactions, bear a large variety of different thioether substituents including linear and branched alkyl groups of C2-C12 length, sugars, aromatic and hydroxyethyl moieties. The cytotoxic/cytostatic activities of these compounds were studied on tumorous SCC and healthy HaCaT cell lines. While nucleosides equipped with bulky substituents such as acetylated pyranoses or a C12 alkyl chain were found to be inactive, compounds bearing an apolar substituents of C3-C6 length displayed potent inhibitory activity on cellular proliferation. Most of the active compounds have higher IC_50_ against the HaCaT cells than against the SCC cells, and the *n*-butyl derivative **10** proved to be the most promising in this aspect. Compound **10** was included in further experiments, namely, determination of cellular generation time, cell size and confluency changes of SCC and HaCaT cell lines upon treatment of **10** by time-lapse imaging video-microscopy. Mother cell size changes suggests a moderate cytotoxic effect on both cell lines. In contrast, cellular generation times were only slightly affected in non-tumorous HaCaT cells, while almost completely blocked in SCC tumor cells. Measurements of cellular proliferation via digital image sequence analyisis of cell monolayer expansion, revealed a proliferative scenescence after the first division of SCC tumor cells in the presence of compound **10**. The changes in the nucleus structure were tested by fluorescent microscopy. Compound **10** essentially affects chromatin structure in a genotoxic manner, specific to SCC tumor cells, resulting a condensed nuclear material and micronucleus induction. Comparing these results with data from the time-lapse microscopy and viability tests, cytostatic effect was most likely observed due to the incorporation of compound **10** during mitosis, rather than direct cytotoxicity.

These results demonstrate, that the thiol-ene reaction of unsaturated nucleosides represents an efficient strategy to produce nucleoside analogue drug candidates with a completely new chemical structure. The cytostatic potency of the 3’-alkylthiomethyl substituted pyrimidine xylofuranosides and the observed remarkable genotoxic effect of the *n*-butylthioether derivative **10** suggest that the C3’-modified xylo-configured nucleosides may be of great value in developing novel anticancer agents.

## 4. Materials and Methods 

### 4.1. General Informations

The 2,2-dimethoxy-2-phenylacetophenone (abbreviated as DMPA or DPAP) and the applied thiols (ethyl-, *n*-propyl-, *n*-butyl-, *n*-hexyl-, *n*-octyl-, *n*-dodecyl-, *i*-propyl-, *t*-butyl-, and *i*-butyl mercaptane) were purchased from Sigma Aldrich Chemical Co. and used without further purification. Compound **1–5** and the exomethylene derivatives were synthetized according to the literature [31]. Optical rotations were measured at room temperature with a Perkin-Elmer 241 automatic polarimeter. TLC was performed on Kieselgel 60 F254 (Merck) with detection by UV-light (254 nm) and immersing into sulfuric acidic ammonium-molibdenate solution or 5% ethanolic sulfuric acid followed by heating. Flash column chromatography was performed on Silica gel 60 (Merck 0.040–0.063 mm). Organic solutions were dried over anhydrous Na_2_SO_4_ or MgSO_4_, and concentrated in vacuum. The ^1^H NMR (360 and 400 MHz) and ^13^C NMR (90 and 100 MHz) spectra were recorded with Bruker DRX-360 and Bruker DRX-400 spectrometers at 25 °C. Chemical shifts are referenced to Me4Si (0.00 ppm for ^1^H) and to the residual solvent signals (CDCl_3_: 77.2, DMSO-d6: 39.5, CD_3_OD: 49.0 for ^13^C). Two-dimensional COSY and ^1^H–^13^C HSQC experiments were used to assist NMR assignments. Copy of NMR spectra of all compounds can be found in Appendix A.

The MALDI-TOF MS measurements were carried out with a Bruker Autoflex Speed mass spectrometer equipped with a time-of-flight (TOF) mass analyzer. In all cases 19 kV (ion source voltage 1) and 16.65 kV (ion source voltage 2) were used. For reflectron mode, 21 kV and 9.55 kV were applied as reflector voltage 1 and reflector voltage 2, respectively. A solid phase laser (355 nm, ≥100 μJ/pulse) operating at 500 Hz was applied to produce laser desorption and 3000 shots were summed. The 2,5-Dihydroxybenzoic acid (DHB) was used as matrix and F_3_CCOONa as cationising agent in DMF. 

ESI-QTOF MS measurements were carried out on a maXis II UHR ESI-QTOF MS instrument (Bruker), in positive ionization mode. The following parameters were applied for the electrospray ion source: Capillary voltage: 3.6 kV; end plate offset: 500 V; nebulizer pressure: 0.5 bar; dry gas temperature: 200 °C and dry gas flow rate: 4.0 L/min. The MS method was tuned according to the examined mass range, which was 200–1000 m/z. Constant background correction was applied for each spectrum, the background was recorded before each sample by injecting the blank sample matrix (solvent). Na-formate calibrant was injected after each sample, which enabled internal calibration during data evaluation. Mass spectra were recorded by otofControl version 4.1 (build: 3.5, Bruker) and processed by Compass DataAnalysis version 4.4 (build: 200.55.2969).

The photoinitiated reactions were carried out in a borosilicate vessel by irradiation with a Hg-lamp giving maximum emission at 365 nm, without any caution to exclude air or moisture. The logP values were calculated using the logP calculation plugin of Marvin Sketch (version 16.5.2) from ChemAxon (Budapest, Hungary) using the Consensus Method with electrolyte concentrations of 0.1 M.

**General method for the low-temperature photoinduced addition of thiols to alkenes**: The set-up consisted of the reaction vessel and the cooling medium (acetone–liquid nitrogen mixture) in a Dewar flask and a UV-lamp placed next to the mixture. For the solution of the starting alkene and thiol in the given solvent, 2,2-dimethoxy-2-phenylacetophenone (DMPA) (0.10 equiv./alkene) was added. The reaction mixture was cooled to the applied temperature, and irradiated with UV light for 15 min. Before irradiation, the entire set-up was covered by an aluminium foil tent. After 15 min DMPA (0.1 equiv.) was added, and the mixture was cooled again and irradiated for another 15 min. The addition of DMPA and irradiation at this temperature was repeated once more. The reaction was monitored by TLC and if the conversion was low, another irradiation cycle were carried out, and if it was necessary, the temperature was increased before the next irradiaton.

### 4.2. Synthesis of Nucleoside Derivatives

**1-[3’-Deoxy-3’-*C*-(*n*-propylsulfanylmethyl)-2’,5’-di-*O*-(*tert*-butyldimethylsilyl)-β-d-xylofuranosyl]-thymine (5).** Compound **7T** (0.41 mmol, 200 mg) and 1-propanethiol (8.0 equiv., 3.3 mmol, 308 µL) and DMPA (10.6 mg, 0.041 mmol, 0.10 equiv.) were dissolved in toluene (1.0 mL) and irradiated at −80 °C for 3× 15 min. The reaction was monitored by TLC (hexane/acetone 85/15 *R*_f_ = 0.42). The solvent was evaporated under reduced pressure. The crude product was purified by flash column chromatography (hexane/acetone 9/1) to give compound **5** (113 mg, 49%) as a white solid. (The d-*xylo*:d-*ribo* ratio was ~ 12.5:1 by the ^1^H-NMR spectrum). [α]_D_: +72.94 (*c* = 0.17, CHCl_3_), R_f_ = 0.42 (85:15 hexane/acetone), ^1^H NMR (400 MHz, CDCl_3_) *δ* (ppm) 8.95 (s, 1H, N*H*), 7.54 (s, 1H, H-6 thymine), 5.94 (d, *J* = 6.9 Hz, 1H, H-1’), 4.31 (d, *J* = 8.2 Hz, 1H, H-4’), 4.13 (dd, *J* = 9.1 Hz, *J* = 7.0 Hz, 1H, H-2’), 4.00 (d, *J* = 11.7 Hz, 1H, H-5’a), 3.89 (dd, *J* = 11.8 Hz, *J* = 2.1 Hz, 1H, H-5’b), 2.85–2.74 (m, 2H, SC*H*_2_), 2.73–2.63 (m, 1H, H-3’), 2.52 (ddd, *J* = 12.2 Hz, *J* = 8.0 Hz, *J* = 4.4 Hz, 2H, SC*H*_2_), 1.96 (s, 3H, C*H*_3_ thymine), 1.68–1.56 (m, 2H, C*H*_2_ propyl), 1.02 (t, *J* = 7.4 Hz, 3H, C*H*_3_ propyl), 0.99, 0.86 (2xs, 18H, 6x*t*-Bu C*H*_3_), 0.18, 0.01, −0.11 (3xs, 12H, C*H*_3_); ^13^C NMR (100 MHz, CDCl_3_) *δ* (ppm) 163.6, 150.8 (2C, 2xthymine *C*O), 135.4 (1C, C-6 thymine), 111.2 (1C, C-5 thymine), 87.3, 78.8, 76.5 (3C, C-1’, C-2’, C-4’), 63.4 (1C, C-5’), 46.5 (1C, C-3’), 34.3 (1C, SCH_2_), 28.6 (1C, S*C*H_2_), 26.1, 25.5 (6C, 2x*t*-Bu *C*H_3_), 22.8 (1C, *C*H_2_ propyl), 18.3, 17.7 (2C, 2x*t*-Bu *C*_q_), 13.4 (1C, *C*H_3_ propyl), 12.3 (1C, *C*H_3_ thymine), −4.6, −4.7, −5.3, −5.4 (4C, 4x*C*H_3_); ESI MS: *m/z* calcd for C_26_H_50_N_2_NaO_5_SSi_2_ [M + Na]^+^ 581.2877, found 581.2866.

**1-[3’-Deoxy-3’-*C*-(ethylsulfanylmethyl)-2’,5’-di-*O*-(*tert*-butyldimethylsilyl)-β-d-xylofuranosyl]-thymine (8)**. Compound **7T** (300 mg, 0.62 mmol), EtSH (367 µL, 4.97 mmol, 8.0 equiv.) and DMPA (15.9 mg, 0.062 mmol, 0.10 equiv.) were dissolved in toluene (1.5 mL) and irradiated at −80 °C for 3× 15 min. The solvent was evaporated, the crude product was purified by flash column chromatography (hexane/acetone 9/1) to give compound (**8**) (290 mg, 86% ~91% xylo ratio) as a white foam. [α]_D_ = +52.63 (c = 0.19 CHCl_3_), R_f_ = 0.36 (hexane/acetone 8/2), ^1^H NMR (360 MHz, CDCl_3_) *δ* (ppm) 9.05 (s, 1H, N*H*), 7.73 (s, 1H, H-6), 6.12 (d, *J* = 6.9 Hz, 1H, H-1’), 4.84–4.70 (m, 1H), 4.48 (d, *J* = 8.1 Hz, 1H), 4.31 (dd, *J* = 9.1, 7.1 Hz, 1H), 4.21–4.13 (m, 1H, H-5’b), 4.07 (dd, *J* = 11.7, 1.6 Hz, 1H, H-5’a), 3.01–2.94 (m, 2H, CH_3_CH_2_SC*H*_2b_ and H-3’), 2.92–2.82 (m, 1H, CH_3_CH_2_SC*H*_2a_), 2.75 (dd, *J* = 14.7, 7.3 Hz, 2H, CH_3_C*H*_2_S), 2.14 (s, 3H, ThymineC*H*_3_), 1.46 (t, *J* = 7.3 Hz, 3H, C*H*_3_CH_2_S), 1.16 (s, 9H, *t*-Bu), 1.04 (s, 9H, *t*-Bu), 0.35 (s, 6H, 2xSiC*H*_3_), 0.19 (s, 3H, SiC*H*_3_), 0.07 (s, 3H, SiC*H*_3_). ^13^C NMR (90 MHz, CDCl_3_) *δ* (ppm) 163.7, 150.9 (2C, 2x*C*O), 135.6 (1C, C-6), 111.3 (1C, C-5), 87.4, 78.9, 76.6 (3C, C-1’, C-2’, C-4’), 63.6 (1C, C-5’), 46.5 (1C, C-3’), 28.3 (2C, 2xS*C*H_2_), 26.3, 25.7 (6C, 2xSiC(*C*H_3_)_3_, 18.4, 17.9 (2C, 2x*t*-Bu*C*_q_), 14.7 (1C, *C*H_3_CH_2_S), 12.5 (1C, Thymine*C*H_3_), −4.5, −4.6, −5.2 (4C, 4xSi*C*H_3_). MS: *m/z* calcd for C_25_H_48_N_2_NaO_5_SSi_2_ [M + Na]^+^ 567.272, found 567.315.

**1-[3’-Deoxy-3’-*C*-(*i*-propylsulfanylmethyl)-2’,5’-di-*O*-(*tert*-butyldimethylsilyl)-β-d-xylofuranosyl]-thymine (9).** Compound **7T** (200 mg, 0.41 mmol), *i*-PrSH (307 µL, 3.31 mmol, 8.0 equiv.) and DMPA (10.6 mg, 0.041 mmol, 0.10 equiv.) were dissolved in THF (1mL) and irradiated at −40 °C for 3× 15 min. After the third irradiation cycle at −40 °C, very low conversion was observed by TLC. Then, the reaction mixture was allowed to warm up to 0 °C and three irradiation cycles were carried out at 0 °C.The solvent was evaporated, the crude product was purified by flash column chromatography (hexane/acetone 9/1) to give compound **9** (77 mg, 34% with 97% diastereomeric purity) as a yellowish syrup. [α]_D_ = +52.9 (*c* = 0.17, CHCl_3_), R_f_ = 0.52 (CH_2_Cl_2_/acetone 95/5), ^1^H NMR (360 MHz, CDCl_3_) *δ* (ppm) 9.96 (s, 1H, N*H*), 7.69 (s, 1H, H-6), 6.10 (d, *J* = 6.8 Hz, 1H, H-1’), 4.44 (d, *J* = 8.2 Hz, 1H, H-4’), 4.28 (dd, *J* = 8.9, 7.3 Hz, 1H, H-2’), 4.14 (d, *J* = 11.5 Hz, 1H, H-5’a), 4.03 (d, *J* = 10.5 Hz, 1H, H-5’b), 3.11 (dt, *J* = 13.3, 6.6 Hz, 1H, *i*-PrC*H*), 2.96 (s, 1H, SC*H*_2_-a), 2.94 (s, 1H, SC*H*_2_-b), 2.82 (td, *J* = 16.3, 8.8 Hz, 1H, H-3’), 2.10 (s, 3H, thymineC*H*_3_), 1.43 (dd, *J* = 6.5, 4.0 Hz, 6H, 2x*i*-PrC*H*_3_), 1.13 (s, 9H, *t*-Bu), 1.00 (s, 9H, *t*-Bu), 0.32 (s, 6H, 2xSiC*H*_3_), 0.16 (s, 3H, SiC*H*_3_), 0.03 (s, 3H, SiC*H*_3_). ^13^C NMR (90 MHz, CDCl_3_) *δ* (ppm) 164.0, 151.1 (2C, 2x*C*O), 135.4 (1C, C-6), 111.3 (1C, C-5), 87.2, 78.8, 76.6 (3C, C-1’, C-2’, C-4’), 63.5 (1C, C-5’), 46.9 (1C, C-3’), 35.1 (1C, *i*-Pr*C*H), 27.2 (1C, S*C*H_2_), 26.2, 25.6 (6C, 2xSiC(*C*H_3_)_3_) 23.3, 23.2 (2C, 2x*i*-Pr*C*H_3_), 18.3, 17.7 (2C, 2x*t*-Bu*C*_q_), 12.4 (1C, thymine*C*H_3_), −4.6, −4.7, −5.3, −5.4 (4C, 4xSi*C*H_3_). ESI MS: *m/z* calcd for C_26_H_50_N_2_NaO_5_SSi_2_ [M + Na]^+^ 581.2877, found 581.2876.

**1-[3’-Deoxy-3’-*C*-(*n*-butylsulfanylmethyl)-2’,5’-di-*O*-(*tert*-butyldimethylsilyl)-β-d-xylofuranosyl]-thymine (10).** Compound **7T** (200 mg, 0.41 mmol), 1-butanethiol (354 µL, 3.31 mmol, 8.0 equiv.) and DMPA (10.6 mg, 0.041 mmol, 0.10 equiv.) were dissolved in THF (1mL) and irradiated at −80 °C for 3× 15 min. After the third irradiation cycle at −80 °C, only ca. 50% conversion was observed by TLC. Then, the reaction mixture was allowed to warm up to −40 °C and three irradiation cycles were carried out at −40 °C. The solvent was evaporated, the crude product was purified by flash column chromatography (hexane/acetone 9/1) to give compound **10** (144 mg, 62%, with ~20:1 xylo:ribo ratio) as a yellowish syrup. The reaction was repeated at 0 °C to give the product a 65% yield in a ~10:1 d-*xylo*:d-*ribo* ratio. [α]_D_ = +38.0 (*c* = 0.10, CDCl_3_), R_f_ = 0.59 (CH_2_Cl_2_/acetone 95/5), ^1^H NMR (360 MHz, CDCl_3_) *δ* (ppm) 9.83 (s, 1H, N*H*), 7.70 (s, 1H, H-6), 6.11 (d, *J* = 6.9 Hz, 1H, H-1’), 4.46 (d, *J* = 8.0 Hz, 1H, H-4’), 4.28 (dd, *J* = 8.8, 7.2 Hz, 1H, H-2’), 4.15 (dd, *J* = 11.5, 0.8 Hz, 1H, H-5’a), 4.04 (dd, *J* = 11.7, 1.5 Hz, 1H, H-5’b), 2.94–2.78 (m, 3H, SC*H*_2_ and H-3’), 2.73–2.66 (m, 2H, SC*H*_2_), 2.11 (s, 3H, thymineC*H*_3_), 1.79–1.68 (m, 2H, BuC*H*_2_), 1.58 (dd, *J* = 14.4, 7.2 Hz, 2H, BuC*H*_2_), 1.14 (s, 9H, *t*-Bu), 1.01 (s, 9H, *t*-Bu), 0.33 (s, 6H, 2xSiC*H*_3_), 0.16 (s, 3H, SiC*H*_3_), 0.04 (s, 3H, SiC*H*_3_). ^13^C NMR (90 MHz, CDCl_3_) *δ* (ppm) 164.0, 151.1 (2C, 2x*C*O), 135.4 (1C, C-6), 111.3 (1C, C-5), 87.2, 78.8, 76.5 (3C, C-1’, C-2’, C-4’), 63.5 (1C, C-5’), 46.5 (1C, C-3’), 31.9, 31.6, 28.6 (3C, 2xS*C*H_2_ and CH_3_CH_2_*C*H_2_CH_2_), 26.2, 25.6 (6C, 2xSiC(*C*H_3_)_3_, 22.0 (1C, CH_3_*C*H_2_CH_2_CH_2_), 18.4, 17.8 (2C, 2x*t*-Bu*C*_q_), 13.7, 12.4 (2C, Bu*C*H_3_ and Thymine*C*H_3_), −4.6, −4.6, −5.2, −5.3 (4C, 4xSi*C*H_3_). ESI MS: *m/z* calcd for C_27_H_52_N_2_NaO_5_SSi_2_ [M + Na]^+^ 595.3033, found 595.3028.

**1-[3’-Deoxy-3’-*C*-(*i-*butylsulfanylmethyl)-2’,5’-di-*O*-(*tert*-butyldimethylsilyl)-β-d-xylofuranosyl]-thymine (11).** Compound **7T** (200 mg, 0.41 mmol), *i*-BuSH (359 µL, 3.31 mmol, 8.0 equiv.) and DMPA (10.6 mg, 0.041 mmol, 0.10 equiv.) were dissolved in toluene (1mL) and irradiated at 0 °C for 3× 15 min. The solvent was evaporated, the crude product was purified by flash column chromatography (gradient elution hexane/acetone 95/5→9/1) to give compound **11** (86 mg, 36% ~22:1 d-*xylo*:d-*ribo* ratio) as a clourless syrup. [α]_D_ = +46 (*c* = 0.10, CHCl_3_), R_f_ = 0.17 (hexane/acetone 8/2), ^1^H NMR (360 MHz, CDCl_3_) *δ* (ppm) 9.42 (s, 1H, N*H*), 7.71 (s, 1H, H-6), 6.12 (d, *J* = 6.9 Hz, 1H, H-1’), 4.48 (d, *J* = 8.0 Hz, 1H, H-4’), 4.29 (dd, *J* = 9.0, 7.0 Hz, 1H, H-2’), 4.17 (d, *J* = 11.6 Hz, 1H, H-5’a), 4.06 (dd, *J* = 11.7, 2.0 Hz, 1H, H-5’b), 3.02–2.79 (m, 4H, H-3’and SC*H*_2_ and *i*-BuC*H*), 2.63–2.55 (m, 2H, SC*H*_2_), 2.13 (s, 3H, ThymineC*H*_3_), 1.18 (d, *J* = 6.7 Hz, 6H, 2x*i*-BuC*H*_3_), 1.16 (s, 9H, *t*-Bu), 1.03 (s, 9H, *t*-Bu), 0.34 (s, 6H, 2xSiC*H*_3_), 0.17 (s, 3H, SiC*H*_3_), 0.06 (s, 3H, SiC*H*_3_). ^13^C NMR (90 MHz, CDCl_3_) *δ* (ppm) 163.9, 151.0 (2C, 2x*C*O), 135.5 (1C, C-6), 111.3 (1C, C-5), 87.3, 78.9, 76.6 (3C, C-1’, C-2’, C-4’), 63.5 (1C, C-5’), 46.7 (1C, C-3’), 41.7 (1C, *i*-Bu*C*H_2_), 29.5 (1C, S*C*H_2_), 28.6 (1C, *i*-Bu*C*H), 26.2, 25.6 (6C, 2xSiC(*C*H_3_)_3_), 22.1, 22.0 (2C, 2x*i*-Bu*C*H_3_), 18.4, 17.8 (2C, 2x*t*-Bu*C*_q_), 12.5 (1C, Thymine*C*H_3_), −4.5, −4.6, −5.2, −5.3 (4C, 4xSi*C*H_3_). ESI MS: *m/z* calcd for C_27_H_52_N_2_NaO_5_SSi_2_ [M + Na]^+^ 595.3033, found 595.3025.

**1-[3’-Deoxy-3’-*C*-(*t*-butylsulfanylmethyl)-2’,5’-di-*O*-(*tert*-butyldimethylsilyl)-β-d-xylofuranosyl]-thymine (12).** Compound **7T** (300 mg, 0.62 mmol), *t*-BuSH (560 µL, 4.97 mmol, 8.0 equiv.) and DMPA (15.9 mg, 0.062 mmol, 0.10 equiv.) were dissolved in THF (1mL) and irradiated at −80 °C for 3× 15 min. After the third irradiation cycle at −80 °C, very low conversion was observed by TLC. Then, the reaction mixture was allowed to warm up to 0 °C and three irradiation cycles were carried out at 0 °C. The solvent was evaporated, the crude product was purified by flash column chromatography (hexane/acetone 9/1) to give an inseparable 14:1 compound **12** (193 mg, 54%, 14:1 d-*xylo*:d-*ribo* ratio) as a colorless syrup. [α]_D_ = +47.1 (*c* = 0.14, CHCl_3_), R_f_ = 0.15 (hexane/acetone 9/1), ^1^H NMR (360 MHz, CDCl_3_) *δ* (ppm) 9.92 (s, 1H, N*H*), 7.70 (s, 1H, H-6), 6.10 (d, *J* = 6.9 Hz, 1H, H-1’), 4.41 (d, *J* = 8.3 Hz, 1H, H-4’), 4.29 (dd, *J* = 9.2, 7.1 Hz, 1H, H-2’), 4.09 (dd, *J* = 11.9, 1.0 Hz, 1H, H-5’a), 4.01 (dd, *J* = 11.8, 1.9 Hz, 1H, H-5’b), 3.01–2.89 (m, 2H, SC*H*_2_), 2.87–2.76 (m, H-3’), 2.10 (s, 3H, thymineC*H*_3_), 1.48 (s, 9H, C(C*H*_3_)_3_), 1.13 (s, 9H, SiC(C*H*_3_)_3_), 1.02 (s, 9H, SiC(C*H*_3_)_3_), 0.32 (s, 6H, 2xSiC*H*_3_), 0.17 (s, 3H, SiC*H*_3_), 0.05 (s, 3H, SiC*H*_3_). ^13^C NMR (90 MHz, CDCl_3_) *δ* (ppm) 164.1, 151.1 (2C, 2x*C*O), 135.4 (1C, C-6), 111.2 (1C, C-5), 87.2 (1C, C-1’), 78.8, 76.7 (2C, C-2’, C-4’), 63.6 (1C, C-6’), 48.0 (1C, C-3’), 42.7 (1C, S*C*H_2_), 30.9 (3C, SC(*C*H_3_)_3_), 26.2, 25.6 (6C, 2xSiC(*C*H_3_)_3)_, 25.3 (1C, S*C*(CH_3_)_3_), 18.3, 17.8 (2C, 2xSi*C*(CH_3_)_3_), 12.4 (1C, thymine*C*H_3_), −4.5, −4.6, −5.3, −5.4 (4C, 4xSi*C*H_3_). ESI MS: *m/z* calcd for C_27_H_52_N_2_NaO_5_SSi_2_ [M + Na]^+^ 595.3033, found 595.3024.

**1-[3’-Deoxy-3’-*C*-(*n*-hexylsulfanylmethyl)-2’,5’-di-*O*-(*tert*-butyldimethylsilyl)-β-d-xylofuranosyl]-thymine (13).** Compound **7T** (200 mg, 0.41 mmol) 1-hexanethiol (470 µL, 3.31 mmol, 8.0 equiv.) and DMPA (10.6 mg, 0.041 mmol, 0.10 equiv.) were dissolved in toluene (1mL) and irradiated at −80 °C for 3× 15 min. After the third irradiation cycle at −80 °C, only ca. 20% conversion was observed by TLC. Then, the reaction mixture was allowed to warm up to −40 °C and three irradiation cycles were carried out at −40 °C. The solvent was evaporated, the crude product was purified by flash column chromatography (gradient elution hexane/acetone 95/5→9/1) to give compound **13** (114 mg, 46%, 30:1 d-*xylo*:d-*ribo* ratio) as a yellowish syrup. [α]_D_ = +41.7 (*c* = 0.12, CHCl_3_), R_f_ = 0.33 (hexane/acetone 9/1), ^1^H NMR (360 MHz, CDCl_3_) *δ* (ppm) 9.68 (s, 1H, N*H*), 7.71 (s, 1H, H-6), 6.12 (d, *J* = 6.9 Hz, 1H, H-1’), 4.46 (d, *J* = 8.1 Hz, 1H, H-4’), 4.29 (dd, *J* = 8.9, 7.1 Hz, 1H, H-2’), 4.16 (d, *J* = 11.2 Hz, 1H, H-5’a), 4.05 (dd, *J* = 11.7, 1.6 Hz, 1H, H-5’b), 2.93 (t, *J* = 10.8 Hz, 2H, SC*H*_2_), 2.89 – 2.79 (m, 1H, H-3’), 2.70 (td, *J* = 7.9, 2.3 Hz, 2H, SC*H*_2_), 2.12 (s, 3H, ThymineC*H*_3_), 1.83–1.68 (m, 3H), 1.60–1.51 (m, 3H), 1.49–1.41 (m, 5H), 1.15 (s, 9H, *t*-Bu), 1.02 (s, 9H, *t*-Bu), 0.34 (s, 6H, 2xSiC*H*_3_), 0.17 (s, 3H, SiC*H*_3_), 0.05 (s, 3H, SiC*H*_3_). ^13^C NMR (90 MHz, CDCl_3_) *δ* (ppm) 164.0, 151.0 (2C, 2x*C*O), 135.5 (1C, C-6), 111.3 (1C, C-5), 87.3, 78.8, 76.5 (3C, C-1’, C-2’, C-4’), 63.5 (1C, C-5’), 46.5 (1C, C-3’), 32.3, 31.5, 29.5, 28.7, 28.6 (5C, 5x*C*H_2_), 26.2, 25.6 (6C, 2xSiC(*C*H_3_)_3_), 22.6 (1C, CH_3_*C*H_2_), 18.4, 17.8 (2C, 2x*t*-Bu*C*_q_), 14.1, 12.4 (2C, Thymine*C*H_3_ and Hexyl*C*H_3_), −4.5, −4.6, −5.2, −5.3 (4C, 4xSi*C*H_3_). ESI MS: *m/z* calcd for C_29_H_56_N_2_NaO_5_SSi_2_ [M + Na]^+^ 623.3346, found 623.3340.

**1-[3’-Deoxy-3’-*C*-(*n*-octylsulfanylmethyl)-2’,5’-di-*O*-(*tert*-butyldimethylsilyl)-β-d-xylofuranosyl]-thymine (14).** Compound **7T** (200 mg, 0.41 mmol), 1-octanethiol (574 µL, 3.31 mmol, 8.0 equiv.) and DMPA (10.6 mg, 0.041 mmol, 0.10 equiv.) were dissolved in toluene (1mL) and irradiated at 0 °C for 3× 15 min. The solvent was evaporated, the crude product was purified by flash column chromatography (gradient elution hexane/acetone 95/5→9/1) to give compound **14** (77 mg, 29%, 24:1 d-*xylo*:d-*ribo* ratio) as a colorless syrup. [α]_D_ = +47.7 (*c* = 0.13, CHCl_3_), R_f_ = 0.34 (hexane/acetone 8/2), ^1^H NMR (360 MHz, CDCl_3_) *δ* (ppm) 9.32 (s, 1H, N*H*), 7.72 (s, 1H, H-6), 6.12 (d, *J* = 6.9 Hz, 1H, H-1’), 4.48 (d, *J* = 8.2 Hz, 1H, H-4’), 4.30 (dd, *J* = 9.2, 7.0 Hz, 1H, H-2’), 4.17 (d, *J* = 10.7 Hz, 1H, H-5’a), 4.06 (dd, *J* = 11.7, 2.0 Hz, 1H, H-5’b), 2.99–2.80 (m, 3H, SC*H*_2_ and H-3’), 2.73–2.67 (m, 2H, SC*H*_2_), 2.13 (s, 3H, thymineC*H*_3_), 1.76 (ddd, *J* = 11.2, 10.7, 5.2 Hz, 2H), 1.61–1.52 (m, 2H), 1.46 (s, 10H), 1.16 (s, 9H, *t*-Bu), 1.03 (s, 9H, *t*-Bu), 0.35 (s, 6H, 2xSiC*H*_3_), 0.18 (s, 3H, SiC*H*_3_), 0.06 (s, 3H, SiC*H*_3_). ^13^C NMR (90 MHz, CDCl_3_) *δ* (ppm) 163.8, 151.0 (2C, 2xThymine*C*O), 135.5 (1C, C-6), 111.3 (1C, C-5), 87.3, 78.9, 76.6 (3C, C-1’, C-2’, C-4’), 63.6 (1C, C-5’), 46.5 (1C, C-3’), 32.3, 31.9, 29.6, 29.3, 29.0, 28.7, 26.3 (7C, 7x*C*H_2_), 25.6 (6C, 2xSiC(*C*H_3_)_3_), 22.7 (1C, OctCH_3_*C*H_2_), 18.4, 17.8 (2C, *t*-Bu*C*_q_), 14.2 (1C, Octyl*C*H_3_), 12.5 (1C, ThymineCH_3_), −4.5, −4.6, −5.2, −5.3 (4C, 4xSi*C*H_3_). ESI MS: *m/z* calcd for C_31_H_60_N_2_NaO_5_SSi_2_ [M + Na]^+^ 651.3659, found 651.3648.

**1-[3’-Deoxy-3’-*C*-(*n*-dodecylsulfanylmethyl)-2’,5’-di-*O*-(*tert*-butyldimethylsilyl)-β-d-xylofuranosyl]-thymine (15).** Compound **7T** (200 mg, 0.41 mmol), 1-dodecanethiol (793 µL, 3.31 mmol, 8.0 equiv.) and DMPA (10.6 mg, 0.041 mmol, 0.10 equiv.) were dissolved in toluene (1mL) and irradiated at 0 °C for 3× 15 min. The solvent was evaporated, the crude product was purified by flash column chromatography (gradient elution hexane/acetone 95/5→9/1) to give compound **15** (130 mg, 46% with 22:1 xylo:ribo ratio) as a colorless syrup. [α]_D_ = +46.2 (*c* = 0.13, CHCl_3_), R_f_ = 0.34 (hexane/acetone 8/2), ^1^H NMR (360 MHz, CDCl_3_) *δ* (ppm) 9.63 (s, 1H, N*H*), 7.71 (s, 1H, H-6), 6.12 (d, *J* = 6.9 Hz, 1H, H-1’), 4.47 (d, *J* = 8.1 Hz, 1H, H-4’), 4.29 (dd, *J* = 9.1, 7.0 Hz, 1H, H-2’), 4.17 (dd, *J* = 11.5, 1.1 Hz, 1H, H-5’a), 4.05 (dd, *J* = 11.7, 2.0 Hz, 1H, H-5’b), 3.00–2.90 (m, 2H), 2.88–2.79 (m, 1H), 2.74–2.63 (m, 2H), 2.12 (s, 3H, ThymineC*H*_3_), 1.44 (s, 20H, 10xC*H*_2_), 1.15 (s, 9H, *t*-Bu), 1.03 (s, 9H, *t*-Bu), 0.34 (s, 6H, 2xSiC*H*_3_), 0.17 (s, 3H, SiC*H*_3_), 0.05 (s, 3H, SiC*H*_3_). ^13^C NMR (90 MHz, CDCl_3_) *δ* (ppm) 163.9, 151.0 (2C, 2x*C*O), 135.5 (1C, C-6), 111.3 (1C, C-5), 87.3, 78.9, 76.6 (3C, C-1’, C-2’, C-4’), 63.5 (1C, C-5’), 46.5 (1C, C-3’), 32.3, 32.0, 29.7, 29.7, 29.6, 29.4, 29.3, 28.9, 28.7 (11C, 11xCH_2_), 26.2, 25.6 (6C, 2xSiC(*C*H_3_)_3_), 22.8 (1C, dodecylCH_3_*C*H_2_), 18.4, 17.8 (2C, 2x*t*-Bu*C*_q_), 14.2 (1C, Dodecyl *C*H_3_), 12.4 (1C, Thymine*C*H_3_), −4.5, −4.6, −5.2, −5.3 (4C, 4xSi*C*H_3_). MS: *m/z* calcd for C_35_H_68_N_2_NaO_5_SSi_2_ [M + Na]^+^ 707.429, found 707.533.

**1-[3’-Deoxy-3’-*C*-(benzylsulfanylmethyl)-2’,5’-di-*O*-(*tert*-butyldimethylsilyl)-β-d-xylofuranosyl]-thymine (16).** Compound **7T** (200 mg, 0.414 mmol), benzyl mercaptane (388 µL, 3.31 mmol, 8.0 equiv.) and DMPA (10.6 mg, 0.0414 mmol, 0.1 equiv.) were dissolved in toluene (1 mL) and irradiated for 4× 15 min at −40 °C. The solvent was evaporated under reduced pressure and the crude product was purified by flash column chromatography (CH_2_Cl_2_/acetone 100/1→98/2) to give compound **16** (207 mg, 82%, 10:1 d-*xylo*:d-*ribo* ratio) as a colorless syrup. [α]_D_ = +44.4 (*c* = 0.16, CHCl_3_), R_f_ = 0.63 (CH_2_Cl_2_/acetone 95/5),^1^H NMR (360 MHz, CDCl_3_) *δ* (ppm) 10.03 (s, 1H, N*H*), 7.69 (s, 1H, H-6), 7.49–7.41 (m, 5H, arom.), 6.10 (d, *J* = 6.9 Hz, 1H, H-1’), 5.46 (s, 1H) 4.43 (d, *J* = 6.7 Hz, 1H, H-2’), 4.26 (t, *J* = 7.6 Hz, 1H, H-4’), 4.17 (d, *J* = 11.9 Hz, 1H, H-5’a), 4.03 (dd, *J* = 11.6, 1.4 Hz, 1H, H-5’b), 3.89 (d, *J* = 6.0 Hz, 2H, BnCH_2_), 2.88–2.75 (m, 3H, H-3’ and SC*H*_2_), 2.11 (s, 3H, thymineC*H*_3_), 1.13 (s, 9H, *t*-Bu), 0.98 (s, 9H, *t*-Bu), 0.31 (s, 3H, SiC*H*_3_), 0.30 (s, 3H, SiC*H*_3_), 0.05 (s, 3H, SiC*H*_3_), 0.01 (s, 3H, SiC*H*_3_). ^13^C NMR (90 MHz, CDCl_3_) *δ* (ppm) 163.9, 151.0 (2C, 2x*C*O), 137.6 (1C, Bn*C*_q_), 135.3 (1C, C-6), 128.6, 127.2 (5C, 5x arom. *C*H), 111.1 (1C, C-5), 87.1, 78.7, 76.4 (3C, C-1’, C-2’, C-4’), 63.5 (1C, C-5’), 45.8 (1C, C-3’), 36.2, 27.8 (2C, 2xS*C*H_2_), 26.1, 25.5 (6C, 2xSiC(*C*H_3_)_3_, 18.2, 17.6 (2C, 2x*t*-Bu*C*_q_), 12.3 (1C, thymine *C*H_3_), −4.8, −5.5 (4C, 4x Si*C*H_3_). MS: *m/z* calcd for C_30_H_50_N_2_NaO_5_SSi_2_ [M + Na]^+^ 629.288, found 629.242.

**1-[3’-Deoxy-3’-*C*-(2-hydroxyethylsulfanylmethyl)-2’,5’-di-*O*-(*tert*-butyldimethylsilyl)-β-d-xylofuranosyl]-thymine (17).** Compound **7T** (100 mg, 0.207 mmol), 2-mercaptoethanol (116 µL, 1.66 mmol, 8.0 equiv.) and DMPA (5.3 mg, 0.0207 mmol, 0.1 equiv.) were dissolved in THF (1 mL) and irradiated for 4× 15 min at −40 °C. The solvent was evaporated under reduced pressure and the crude product was purified by flash column chromatography (hexane/acetone 95/5→9/1→8/2) to give compound **17** (83 mg, 72%) as a colorless syrup (4:1 d-ribo: d-xylo ratio). The reaction was repeated at −80 °C, irradiated for 6× 15 min, with 75% yield and ~3:1 xylo:ribo ratio. The reaction w as repeated at 0 °C, irradiated for 4× 15 min, with 74% adn 3.8:1 d-*xylo*:d-*ribo* ratio. [α]_D_ = +45.5 (*c* = 0.11, CHCl_3_), R_f_ = 0.31 (hexane/acetone 8/2), ^1^H NMR (360 MHz, CDCl_3_) *δ* (ppm) 9.59 (s, 1H, N*H*), 7.69 (s, 1H, H-6), 6.09 (d, *J* = 6.9 Hz, 1H, H-1’), 4.48 (d, *J* = 8.3 Hz, 1H, H-4’), 4.34–4.21 (m, 2H), 4.15 (dd, *J* = 11.7, 1.2 Hz, 1H, H-5’a), 4.05 (dd, *J* = 11.9, 2.3 Hz, 1H, H-5’b), 3.98–3.86 (m, 3H), 3.02–2.79 (m, 6H), 2.11 (s, 3H, thymineC*H*_3_), 1.14 (s, 9H, *t*-Bu), 1.02 (s, 9H, *t*-Bu), 0.34 (s, 6H, 2xSiC*H*_3_), 0.16 (s, 3H, SiC*H*_3_), 0.05 (s, 3H, SiC*H*_3_). ^13^C NMR (90 MHz, CDCl_3_) *δ* (ppm) 163.9, 151.0 (2C, 2x*C*O), 135.5 (1C, C-6), 111.3 (1C, C-5), 87.4, 78.7, 76.5 (3C, C-1’, C-2’, C-4’), 63.5, 60.5 (2C, C-5’ and HO*C*H_2_), 46.7 (1C, C-3’), 35.3, 28.7 (2C, 2xSCH_2_), 26.2, 25.6 (6C, 2xSiC(*C*H_3_)_3_), 18.4, 17.8 (2C, 2x*t*-Bu*C*_q_), 12.4 (1C, thymine*C*H_3_), −4.6, −5.3 (4C, 4x Si*C*H_3_). MS: *m/z* calcd for C_25_H_48_N_2_NaO_6_SSi_2_ [M + Na]^+^ 583.267, found 583.197.

**1-[3’-Deoxy-3’-*C*-(*n*-butylsulfanylmethyl)-2’,5’-di-*O*-(*tert*-butyldimethylsilyl)-β-d-xylofuranosyl]-uracil (18).** Compound **7U** (194 mg, 0.414 mmol), 1-butanethiol (354 µL, 3.31 mmol, 8.0 equiv.) and DMPA (10.6 mg, 0.0414 mmol, 0.1 equiv.) were dissolved in THF (1 mL) and irradiated for 6× 15 min at −40 °C. The solvent was evaporated under reduced pressure and the crude product was purified by flash column chromatography (hexane/acetone 95/5→9/1) to give compound **18** (135 mg, 59%) as a colorless syrup (d-*xylo*:d-*ribo* ratio is ~ 60:1). The reaction were repeated at 0 °C to give the product with 66 % yield and with a ~12:1 d-*xylo*:d-*ribo* ratio. [α]_D_ = +65.7 (*c* = 0.21, CHCl_3_), R_f_ = 0.41 (hexane/acetone 8/2),^1^H NMR (360 MHz, CDCl_3_) *δ* (ppm) 9.94 (s, 1H, N*H*), 8.15 (d, *J* = 8.1 Hz, 1H, H-6), 6.13 (d, *J* = 6.6 Hz, 1H, H-1’), 5.89 (dd, *J* = 8.1, 1.8 Hz, 1H, H-5), 4.47 (d, *J* = 6.9 Hz, 1H), 4.29 (dd, *J* = 8.1, 7.1 Hz, 1H), 4.14 (d, *J* = 11.5 Hz, 1H, H-5’a), 4.04 (dd, *J* = 11.8, 1.8 Hz, 1H, H-5’b), 2.95–2.79 (m, 3H, H-3’ and C*H*_2_), 2.75–2.63 (m, 2H, C*H*_2_), 1.80–1.65 (m, 2H, C*H*_2_), 1.63–1.52 (m, 2H, C*H*_2_), 1.11 (s, 9H, *t*-Bu), 1.02 (s, 9H, *t*-Bu), 0.30 (s, 6H, 2xSiC*H*_3_), 0.16 (s, 3H, SiC*H*_3_), 0.05 (s, 3H, SiC*H*_3_). ^13^C NMR (90 MHz, CDCl_3_) *δ* (ppm) 163.5, 150.9 (2C, 2x*C*O), 140.5 (1C, C-6), 102.9 (1C, C-5), 87.7, 79.3, 77.4 (3C, C-1’, C-2’, C-4’), 63.6 (1C, C-5’), 46.6 (1C, C-3’), 31.9, 31.6, 28.8 (3C, 3x*C*H_2_), 26.0, 25.6 (6C, 2xSiC(CH_3_)_3_), 22.0 (1C, Bu*C*H_2_), 18.3, 17.7 (2C, 2x*t*-Bu*C*_q_), 13.7 (1C, butyl *C*H_3_), −4.6, −4.7, −5.4, −5.6 (4C, 4x Si*C*H_3_). MS: *m/z* calcd for C_26_H_50_N_2_NaO_5_SSi_2_ [M + Na]^+^ 581.288, found 581.230. 

**1-[3’-Deoxy-3’-*C*-(*n*-ethylsulfanylmethyl)-β-d-xylofuranosyl]-thymine (19).** Compound **8** (231 mg, 0.4239326 mmol) was dissolved in dry THF (1 mL) and tetrabutylammonium fluoride (TBAF) (1.05 mL from 1M solution in hexane, 1.05 mmol, 2.5 equiv.) was added and stirred for 1 h at r.t. After 1 h, since the conversion was low, TBAF (1.0 equiv.) was added and stirred for 2 h. After 2 h, another portion of TBAF (1.0 equiv.) was added and stirred overnight. Next day, the solvent was evaporated under educed pressure and the crude product was purified by flash column chromatography (EtOAc/MeOH 98/2) to give compound **19** (40 mg, 30%) as a white solid. [α]_D_ = +57.3 (*c* = 0.11, CHCl_3_), R_f_ = 0.51 (EtOAc/MeOH 95/5), ^1^H NMR (360 MHz, DMSO) *δ* (ppm) 11.31 (s, 1H, N*H*), 7.92 (s, 1H, H-6), 5.71 (d, *J* = 7.0 Hz, 1H, H-1’), 5.51 (d, *J* = 5.7 Hz, 1H, 2’O*H*), 5.26 (t, *J* = 3.8 Hz, 1H, 5’O*H*), 4.14 (d, *J* = 8.1 Hz, 1H), 4.00 (dd, *J* = 16.2, 6.7 Hz, 1H), 3.66 (q, *J* = 12.4 Hz, 2H, H-5’ab), 2.85–2.70 (m, 2H, SC*H*_2_), 2.60–2.52 (m, 2H, SC*H*_2_), 1.78 (s, 3H, thymineC*H*_3_), 1.20 (t, *J* = 7.4 Hz, 3H, SCH_2_C*H*_3_). ^13^C NMR (90 MHz, DMSO) *δ* (ppm) 163.7, 151.1 (2C, 2x*C*O), 136.6 (1C, C-6), 109.5 (1C, C-5), 87.4 (1C, C-1’), 79.0, 75.4 (2C, C-2’, C-4’), 60.8 (1C, C-5’), 45.6 (1C, C-3’), 28.1, 25.3 (2C, 2xS*C*H_2_), 14.6 (1C, SCH_2_*C*H_3_), 12.3 (1C, thymine*C*H_3_). MS: *m/z* calcd for C_13_H_20_N_2_NaO_5_S [M + Na]^+^ 339.099, found 339.236.

### 4.3. Cell Lines and Cell Culture Conditions

The HaCaT cell line was derived from human skin, the cells spontaneously transformed in vitro during long time incubation [47,48]. Cells were cultured in DMEM medium (Biocenter, Szeged, Hungary) containing 10% fetal bovine serum (FBS, Hyclon, Logan, UT) and 1% antibiotic-antimycotic mix (Penicillin–Streptomycin–Neomycin) [49].

SCC-VII is a carcinoma tumor that originally arose in the abdominal wall of C3H mice in Dr. Herman Suit’s laboratory at Harvard University, Boston, Massachusetts. SCC VII cells were grown in Dulbecco’s Modified Eagle’s Medium Nutrient Mixture (DMEM-HAM’S F12) (Sigma-Aldrich) supplemented with 2 mM L-glutamine, 23 mM NaHCO_3_, 100 U/mL penicillin, 100 U/mL streptomycin 1% non-essential amino acids and 10% fetal bovine serum [50].

Cell cultures had been kept under the same conditions during the experiments. Cells were stored at 37 °C, 95% humidity, and 5% CO_2_ in an incubator. During the MTT-assay, different starter cell number was used because of the cell cycle differences; HaCaT was started with 10.000/well meanwhile SCC VII with 5000/well. After passage, 24 h was assured for the cells to the adhesion. Then incubated with the compounds for 48 h. The experiments were performed in 96 well plates.

In LTS video-microscopy studies, ~500.000/350.000 starting cell numbers (T-25 cell culture flask) had been used. Cell cultures had been treated with compound **10** using the IC_50_ concentration corresponding to the SCC VII.

### 4.4. MTT Assay

For the dilution of different concentrations of nucleoside analogues were prepared with DMSO 1% (*v*/*v*) and DMEM / DMEM-F12/Ham’s F12 cell culture medium.

After three days of incubation in the cell culture flask, cells were trypsinised into single cell suspensions. Aliquots (200 μL) of each cell suspension were placed in wells of 96-well-plates (Biocenter, Szeged, Hungary). Preliminary tests were conducted with cell numbers 5000 (SCC-VII) and 10,000 (HaCaT) per well. We have been tested five concentrations of each nucleoside analogues (20-10-5-2.5-1.25 µg/mL). The nucleoside analogues were dissolved in DMSO (1 *v*/*v*%) and DMEM/ DMEM-F12/Ham’s F12 were added to the wells to produce the required concentrations and the cells were similarly incubated for a further 48 h.

MTT (Sigma Aldrich, Budapest, Hungary) solution (0.5 mg/mL in PBS) 100 µL was added to each well. The plates were incubated for 1–2 h at 37 ^◦^C, the wells aspirated, and MTT formazan extracted with 100 µL of DMSO aided by gentle agitation on a shaker. After 10 min at room temperature, the absorbances were read at 570 nm by an automatic plate reader (WALLAC Victor 2 1420 spectrophotometer), the instrument having been blanked beforehand on a row to which cells had not been added. Percentage viability (respiratory competence) of the population of cells in each well was expressed as: (Absorbance of treated cells/Absorbance of control cells) × 100. IC_50_ values of the nucleoside analogues were determined by Graphpad.

### 4.5. Time-Lapse Image Video-Microscopy and Image Analysis

TLS system architecture: Using the TLS system, various parameters had been tested. Control, DMSO control, and IC_50_ concentration of compound **10** compared to the above. Based on literature data, 1% (*v*/*v*) is the amount that is tolerated even without significant change in cells. However, this may vary depending on the sensitivity of the cell [51,52].

Imaging setup:Inverse microscopes sitting in the incubator CO_2_ (SANYOMCO18-AC, Wood Dale, USA).Illumination under minimized heat- and phototoxicity, operated in the near-infrared range (940 nM), with light emitting diodes synchronized by short (1s) image-acquisition periods. Light intensity/energy was limited to the lowest possible level for image acquisition. Cells were only illuminated during image acquisition periods.Cell culture: The growth of cells started in 25 mL T-flasks at medium ~30–50 % confluency [53,54].

National Health Institute’s ImageJ software was used to analyze the image sequences of the time-lapse videomicroscopy [55].

The image analyzing method included:

Mother cells size measurement:Opening the image sequence in 8-bit format.Stack Deflicker: The Stack Deflicker calculates the average grey value for each frame and normalizes all frames so that they have the same average grey level as a specified frame of the stack. This plugin is very useful to remove flickering in movies caused by frame rates different from the frequency of AC used for the light-source that illuminate the scene. An input value of −1 corresponds to the brightest frame while an input value of zero corresponds to the faintest frame. If a region of the stack is selected the average frame intensity will be calculated from this region [56].Changing the brightness/contrast, if it is neccesary.Subtrack background: Removes smooth continuous backgrounds from gels and other images. Based on the concept of the ‘rolling ball’ algorithm described by Sternberg Stanley. Imagine that the 2D grayscale image has a third dimension (height) by the image value at every point in the image, creating a surface. A ball of given radius is rolled over the bottom side of this surface; the hull of the volume reachable by the ball is the background to be subtracted [57].Treshold: Use this tool to automatically or interactively set lower and upper threshold values, segmenting grayscale images into features of interest and background [58].Divided cells were selected on the binary image sequence based on their circularity determined by area/perimeter ratio.Bigger pre-division mother cells were separated from smaller post-division daughter cells.Pixel size was calibrated with Burker chamber.Area calculation from pixel^2^ to µm^2^.

#### 4.5.1. Determination of generation time

Images were acquired at a frame rate of 1 frame per minute. Two daughter cells of a dividing mother cell were followed in time until their next division, resulting in 2 pair of new daughter cells. 

#### 4.5.2. Confluence

Cellular monolayer proliferation was determined by separating the cells as foreground, and breeding surface as background. After the segmentation (treshold) total cell-covered surface was calculated for each image in the 48 h sequence. The data were exported to Graphpad (Prism).

#### 4.5.3. Statistical analysis

Statistical analyses were performed using Prism software (GraphPad), one-way ANOVA for multiple data sets were applied, P < 0.05 was considered statistically significant.

#### 4.5.4. Fluorescent microscopy

For fluorescence staining, cells were breeded in the ibidi microscope slide chamber. The sample preparation process was similar to that of TLS studies, except that after 48 h the samples were fixed with PFA and washed with PBS and left to dry at room temperature overnight. DAPI dye was used for fluorescencnt labbelling of A-T rich regions of DNA.

Dehydrated slides containing fixated cells were mounted in 35 μL DAPI-Antifade Medium under 24 × 50 mM coverslips. Fluorescence of DAPI (4′,6-diamidino-2-phenylindole) was monitored by fluorescence microscopy (Carl Zeiss Compound Universal Microscope III RS).

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
