# Peer review of "Synthesis and Cytostatic Effect of 3’-deoxy-3’-C-Sulfanylmethyl Nucleoside Derivatives with d-xylo Configuration"

_molecules, 2019, doi:10.3390/molecules24112173_

Round 1
Reviewer 1 Report
The authors describe the synthesis of a family of sulphur containing xylofuranose-pyrimidine nucleoside analogues. The substrates are prepared in satisfactory diastereomeric ratios and, in most cases, with good yields by an oxidation-methylenation protocol of the parent hydroxy nucleosides. Taking the advantage if this method, the authors report several nucleoside derivatives bearing a sulfanylmethyl group at the 3’ position with different protecting groups. Moreover, those substrates are tested against SCC-VII cell lines, using HaCaT cell line as control experiment. The work is clearly presented and the experimental procedures and characterization of the compounds are carefully detailed. However, the cytotoxic effect of the reported nucleosides is not very impressive and their selectivity respect to healthy cells is modest. Moreover, the authors provide some further experiments for n-butyl substituted derivative such as the determination of cellular generation time, cell size and confluence changes of both cell lines using time-lapse imaging video-microscopy that demonstrate their potential as anticancer agents. The authors justify the lack of cytotoxic effect on compounds bearing acetylated pyranoses or dodecyl groups at the sulphur atom in the bulkiness of the substituent that, in my opinion, may not be real. If that is true I would not expect such a big difference between n-octyl and n-dodecyl derivatives and, on the contrary , I would expect a decreased cytotoxic effect for branched substituents like i-propyl, i-butyl and, especially, t-butyl. Other thing that surprises me is that the authors did not test different substituents in the hydroxyl groups. The fact that substrate 19, obtained by the deprotection of TBDMS substituted nucleoside, did not show any cytotoxic activity makes obvious the relevance of the substitution in those positions into the activity of the substrates. The evaluation of the most promising n-butyl substituted substrate bearing different substituents at the hydroxyl groups is advisable. Overall, I think they authors provide a nice contribution to the fields of the synthesis of nucleoside derivatives and chemotherapeutic reagents that will be of interest for many organic and medicinal chemists and, consequently I support the publication of the manuscript in Molecules after the comments above have been taken into account.
I also think that a fine revision of the text is advisable. I could find some typos or wrong formatting (i.e. Line 50 “which” not “wich”, Line 107 “branched” not “branched”, Line 167 “concentration” not “concentracion”, Line 225 “Parallel” not “Paralell”, Line 249 “cell-cycle” not “cell-cyle”, Lines 374, 391, 406, 423, 442, 457, 474, 492, 508, “were” not “was”...)
Author Response
Thank you very much for constructive comments which helped us in improving the manuscript. Our responses to the comments are explained in detail below:
„The authors justify the lack of cytotoxic effect on compounds bearing acetylated pyranoses or dodecyl groups at the sulphur atom in the bulkiness of the substituent that, in my opinion, may not be real. If that is true I would not expect such a big difference between n-octyl and n-dodecyl derivatives and, on the contrary , I would expect a decreased cytotoxic effect for branched substituents like i-propyl, i-butyl and, especially, t-butyl.”
Answer:
We agree with this comment. The sentence about the bulkiness (" We assumed that the inactivity of the sugar-substituted 1, 3 and 4 and the n-dodecyl-bearing 15 can be explained by the bulkiness of the 3’-substituents.”) has been deleted from the manuscript.
Other thing that surprises me is that the authors did not test different substituents in the hydroxyl groups. The fact that substrate 19, obtained by the deprotection of TBDMS substituted nucleoside, did not show any cytotoxic activity makes obvious the relevance of the substitution in those positions into the activity of the substrates. The evaluation of the most promising n-butyl substituted substrate bearing different substituents at the hydroxyl groups is advisable.
Answer: Thank you very much for the suggestion. We completely agree with the opinion, that the substituents at 2’ and 5’ positions play crucial role in the cytotoxic activity. We intend to study the effects of substituents ont he bioactivity and have already started the reproduction of the most promising compounds. This work is under way.
I also think that a fine revision of the text is advisable. I could find some typos or wrong formatting (i.e. Line 50 “which” not “wich”, Line 107 “branched” not “branched”, Line 167 “concentration” not “concentracion”, Line 225 “Parallel” not “Paralell”, Line 249 “cell-cycle” not “cell-cyle”, Lines 374, 391, 406, 423, 442, 457, 474, 492, 508, “were” not “was”...)
Answer: Thank you very much for drawing our attention to typos. We attempted to correct all typos in the text.
Reviewer 2 Report
In this manuscript, the authors reported the synthesis of 3’-deoxyl-3’-C-sulfanylmethyl nucleoside derivatives and evaluated their cytostatic effect on two cell lines. But,
1). All the approved or clinically investigated nucleoside analogues are polar and hydrophilic, could you explain, why in your system, polar nucleoside (like compound 19) doesn’t show any activity, while, hydrophobic nucleosides with protection groups have activity. The authors just cited the IC50 of 5-FU from ref 24 as positive control, why is it not re-evaluated on SCC IV cell lines which are not completely same with those in ref 24.
2) Table 3, the unit of IC50 is missing.
3)Table 3, 4, and 5, all data should keep same numbers of significant digits. Table 1, the legend is too easy, reaction condition in details should be added.
4) The writing is good but need to be checked carefully, some typos were found, such as, line 50 “in wich”
In short, I think, this manuscript could be considered for publication on Molecules after minor revisions.
Author Response
Thank you very much for constructive comments which helped us in improving the manuscript. Our responses to the comments are explained in detail below:
1). All the approved or clinically investigated nucleoside analogues are polar and hydrophilic, could you explain, why in your system, polar nucleoside (like compound 19) doesn’t show any activity, while, hydrophobic nucleosides with protection groups have activity. The authors just cited the IC50 of 5-FU from ref 24 as positive control, why is it not re-evaluated on SCC IV cell lines which are not completely same with those in ref 24.
Answer: Indeed, it is surprising that the the unprotected compound 19 was inactive, while some of the the protected derivatives showed promising actvity. However, this finding is not unprecedented. Moreover, there are numerous examples in the literature for the positive effect of silyl substituents on cytotoxic or antiviral activity of nucleoside analogues (see Ref 27-32 in the Reference list). We discussed the potential explanations in detail in the manuscript (pages 8-9, lines 194-214).
In the cell viability test, methotrexate was used as a positive control. The measured IC50 values of methotrexate have been added to Table 3. As methotrexate is not a nucleoside analogue we also added the literature IC50 of 5-FU to the Table 3. We agree that it is not completely accurate. We will purchase 5-FU and in future studies use it as a positive control.
2) Table 3, the unit of IC50 is missing.
Answer: The unit of IC50 (microM) is given in the footnote of Table 3.
3)Table 3, 4, and 5, all data should keep same numbers of significant digits. Table 1, the legend is too easy, reaction condition in details should be added.
Answer: The data of Tables 3-5 have beeen corrected. The reaction details (solvent, reaction time, amount of catalyst) have been added to the legend of Table 1.
4) The writing is good but need to be checked carefully, some typos were found, such as, line 50 “in wich”.
Answer: Thank you for the comment. Typos have been corrected in the text.
Reviewer 3 Report
In the paper entitled "Synthesis and cytostatic effect of 3’-deoxy-3’-C-sulfanylmethyl nucleoside derivatives with D-xylo configuration"the authors present the synthesis of some 3'-deoxy-3'C-thiomethyl nucleoside derivatives via thiol-ene click chemistry. The obtained compounds were investigated for their in vitro antiproliferative and cytotoxic effects. The results obtained suggested that some of these compounds have a slight selectivity towards tumorous cells, and the presence of the silyl protecting group and that of apolar substituents of C3-C6 lenght is essential for their biologic activity. In my opinion, the paper is very well written in English, the hypothesis and the aims are very clear, and the results are also very clearly presented. Therefore, it is suitable for publicatication in Molecules, after some minor changes:
2,2-dimethoxy-2-phenylacetophenone should be abbreviated as DMPA;
The conclusion part should be pointed out;
page 14-line 370: replace "irradtiatons" with "irradiations";
in vitro should be marked in Italic.
Author Response
Thank you very much for constructive comments which helped us in improving the manuscript. Our responses to the comments are explained in detail below:
In my opinion, the paper is very well written in English, the hypothesis and the aims are very clear, and the results are also very clearly presented. Therefore, it is suitable for publicatication in Molecules, after some minor changes:
2,2-dimethoxy-2-phenylacetophenone should be abbreviated as DMPA;
Answer: It has been corrected.
The conclusion part should be pointed out;
Answer: The conclusion part has been pointed out as a subchapter on Page 12
page 14-line 370: replace "irradtiatons" with "irradiations";
in vitro should be marked in Italic.
Answer: We made these corrections.